# Systemic Autoimmune Diseases in Patients Hospitalized with COVID-19 in Spain: A Nation-Wide Registry Study

**DOI:** 10.3390/v14081631

**Published:** 2022-07-26

**Authors:** Víctor Moreno-Torres, Carmen de Mendoza, Susana Mellor-Pita, María Martínez-Urbistondo, Pedro Durán-del Campo, Pablo Tutor-Ureta, José-Manuel Vázquez-Comendador, Jorge Calderón-Parra, Elena Múñez-Rubio, Antonio Ramos-Martínez, Ana Fernández-Cruz, Raquel Castejón, Juan-Antonio Vargas-Nuñez

**Affiliations:** 1Internal Medicine Department, Hospital Universitario Puerta de Hierro-Majadahonda, 28660 Madrid, Spain; cmendoza.cdm@gmail.com (C.d.M.); susanamellor@hotmail.com (S.M.-P.); mmurbistondo@gmail.com (M.M.-U.); pedrodurandc@hotmail.com (P.D.-d.C.); pablo.tutor@hotmail.com (P.T.-U.); koldovazquezcom@gmail.com (J.-M.V.-C.); jorge050390@gmail.com (J.C.-P.); elmuru@gmail.com (E.M.-R.); aramos220@gmail.com (A.R.-M.); anafcruz999@gmail.com (A.F.-C.); raquel.castejon@salud.madrid.org (R.C.); juanantonio.vargas@salud.madrid.org (J.-A.V.-N.); 2Pharmaceutical and Health Sciences, University CEU-San Pablo, 28003 Madrid, Spain; 3Medicine Department, School of Medicine, Universidad Autónoma de Madrid, 28029 Madrid, Spain

**Keywords:** COVID-19, SARS-CoV-2, Systemic Autoimmune Diseases, mortality, Sjogren’s Syndrome, Systemic Vasculitides, Systemic Lupus Erythematosus, Sarcoidosis, Systemic Sclerosis, Mixed and Undifferentiated Connective Tissue Disease, Behçet’s Disease, Inflammatory Myopathies

## Abstract

We aimed to evaluate the clinical outcome of Systemic Autoimmune Diseases (SADs) patients hospitalized with COVID-19 in Spain, before the introduction of SARS-CoV-2 vaccines. A nationwide, retrospective and observational analysis of the patients admitted during 2020, based on the ICD10 codes in the National Registry of Hospital Discharges, was performed. Among 117,694 patients, only 892 (0.8%) presented any type of SAD before COVID-19-related admission: Sjogren’s Syndrome constituted 25%, Systemic Vasculitides 21%, Systemic Lupus Erythematosus 19%, Sarcoidosis 17%, Systemic Sclerosis 11%, Mixed and Undifferentiated Connective Tissue Disease 4%, Behçet’s Disease 4% and Inflammatory Myopathies 2%. The in-hospital mortality rate was higher in SAD individuals (20% vs. 16%, *p* < 0.001). After adjustment by baseline conditions, SADs were not associated with a higher mortality risk (OR = 0.93, 95% CI 0.78–1.11). Mortality in the SADs patients was determined by age (OR = 1.05, 95% CI 1.04–1.07), heart failure (OR = 1.67, 95% CI 1.10–2.49), chronic kidney disease (OR = 1.29, 95% CI 1.05–1.59) and liver disease (OR = 1.97, 95% CI 1.13–3.44). In conclusion, the higher COVID-19 mortality rate seen in SADs patients hospitalized in Spain in 2020 was related to the higher burden of comorbidities, secondary to direct organ damage and sequelae of their condition. Whilst further studies should evaluate the impact of baseline immunosuppression on COVID-19 outcomes in this population, efforts should be focused on the optimal management of SAD to minimize the impact of the organ damage that has been shown to determine COVID-19 prognosis.

## 1. Introduction

The surge of SARS-CoV-2 infections in China towards the end of 2019 changed the global context in an unprecedented way. In the two years following the onset of the pandemic, more than 18 million deaths have been confirmed globally [1]. Fortunately, the experience accumulated during these two years has allowed an understanding of the wide clinical spectrum and heterogeneity of the disease. The importance of identifying populations at risk, and the subsequent need for individualizing medical care and surveillance, have been highlighted [2].

Certain conditions, including age, hypertension, obesity, chronic kidney disease or heart failure, among others, were soon revealed as predictors of COVID-19 severity [3,4]. However, the role of immunosuppression has been more controversial [5,6,7]. The proven efficacy of certain immunosuppressants, such as glucocorticoids and IL-6 inhibitors (e.g., tocilizumab), especially during the hyperinflammatory phase of the disease, suggests that immunosuppression could provide protection against COVID-19 [8,9,10]. In parallel, other authors identified better outcomes in the immunocompromised population and proposed a possible protecting role of a weaker immune response [5]. By contrast, other cohorts have indeed confirmed that baseline immunodeficiencies are associated with worse prognosis [6,7]. Taken together, it seems that the distinct etiologies of impaired immunity and the drugs prescribed for treatment might explain these differences in COVID-19 outcomes [7,11].

Systemic Autoimmune Diseases (SADs) are comprised of a group of heterogeneous disorders in which persistent immune activation leads to a chronic inflammatory environment and direct tissue injury [12]. Consequently, attenuating this uncontrolled immune response with immunosuppressants is the cornerstone of their management. In addition, previous studies have shown that patients with SADs present an impaired effective immune response towards certain agents, including viruses [13]. Therefore, COVID-19 prognosis in this population might be determined by organ damage, immunosuppression and the underlying autoimmune disease itself.

In light of these considerations, the aim of the present study was to evaluate the clinical outcomes of SADs patients hospitalized with COVID-19 in Spain, before the introduction of SARS-CoV-2 vaccines.

## 2. Materials and Methods

A retrospective study with data from population-based hospital discharge diagnoses at the Minimum Basic Data Set (MBDS) of the Spanish National Registry of Hospital Discharges (SNRHD) was performed. The SNRHD is a national public registry that belongs to the Spanish Ministry of Health. It records information from all patients discharged at hospitals/clinics across the country since the nineties [14]. Prior studies have been performed using this registry for other illnesses, including infectious diseases and SADs, and have demonstrated its high value for producing estimates of current burden and time trends for different clinical conditions at a national level [15,16].

### 2.1. Study Population

Our study was performed using all the data from 1 January to 31 December 2020 included in the SNRHD. The criteria for diseases and procedures were defined according to the International Classification of Diseases-10th Revision, Clinical Modification (ICD-10-CM). We selected all hospital admissions assigned with the code U07.1 (COVID-19) as the main diagnosis.

Data regarding demographics and outcomes, including age, gender, ethnicity, length of admission, intensive care unit (ICU) admission or death were retrieved from the database. Baseline conditions, as well as the presence of respiratory insufficiency, were recorded from other ICD-10 codes in the data set, regardless of position, for each episode of hospital admission. In addition, the Charlson Comorbidity Index (CCI), which is a well-validated composite that predicts clinical outcomes in multiple illnesses, was calculated from the previous data [17]. Among other medical conditions, it included diabetes, heart failure, dementia, chronic kidney disease, liver disease and cancer, most of which have been associated with severe COVID-19 [3,4,6]. However, we excluded connective tissue disease from this index in order to analyze SAD as a separate variable.

Patients were labelled as presenting SADs before COVID-19 related admission if any of the following diagnoses were identified: Systemic Lupus Erythematosus (SLE), Sarcoidosis, Sjogren’s Syndrome (SjS), Systemic Sclerosis (SSc), Idiopathic Inflammatory Myopathies (IIM), including Dermatomyositis, Polymyositis and other Inflammatory or Immune Myopathies, Mixed and Undifferentiated Connective Tissue Disease (MUCTD), Behçet’s disease (BD) and Systemic Vasculitides (SVs), such as Giant Cell Arteritis, Takayasu’s Disease, Polyarteritis Nodosa and ANCA-Associated Vasculitides. Other rheumatic diseases or chronic inflammatory arthritis, such as Rheumatoid Arthritis (RA), Polymyalgia Rheumatica (PM), Psoriatic Arthritis (PsA) or other Spondylarthritis were not considered to be SADs and, therefore, were not analyzed.

### 2.2. Statistical Analysis

Categorical variables were reported as frequencies and percentages while continuous variables were presented by their mean and standard deviation. The significance of baseline differences and outcomes of SADs patients and non-SADs patients were determined by the Chi-square, Fisher’s or Student’s t-test, as appropriate. Multivariate logistic regression analyses were performed to determine the factors related to in-hospital mortality and the role of SADs, considering male sex, high blood pressure, obesity, CCI and respiratory insufficiency. We analyzed the overall SAD patient population and performed stratified analyses for each individual SAD as well. Finally, the main prognostic comorbidities for SAD patients were determined. For all the analyses, a significance level of 0.05 was set. Statistical analysis was performed using SPSS v. 26.0 (IBM, Madrid, Spain) software.

## 3. Results

### 3.1. Population Characteristics

A total of 117,694 adults were hospitalized in Spain with COVID-19 during 2020. Overall, 66,685 patients were male (57%) with a mean age at admission of 66.5 years (Table 1). Only 892 patients of these 117,694 patients (0.8%) presented any SAD: 225 (25%) SjS, 185 (21%) SV, 173 (19%) SLE, 149 (17%) Sarcoidosis, 100 (11%) SSc, 39 (4%) MUCTD, 36 (4%) BD and 21 (2%) IIM. Patients with SADs were less frequently male (29% vs. 57%, *p* < 0.001) and Latin-American (6% vs. 8%, *p* = 0.004) than non-SADs patients. No other differences were found considering age and other ethnic groups.

Table 2 gives the distribution of major baseline comorbidities for COVID-19 mortality in the study population. Several differences were found between the SADs and non-SADs groups. Certain conditions, such as heart failure (18% vs. 12%, *p* < 0.001), peripheral vascular disease (8% vs. 4%, *p* < 0.001), chronic lung disease (23% vs. 14%, *p* < 0.001), liver disease (8% vs. 5%, *p* < 0.001) and chronic kidney disease (18% vs. 11%, *p* < 0.001), were significantly more frequent among SADs patients, while diabetes was more frequent in the non-SADs group (24% vs. 21%, *p* = 0.023). Accordingly, the CCI was significantly higher in SADs patients (3.9 vs. 3.5, *p* < 0.001).

### 3.2. Outcomes and Mortality Risk

The clinical outcomes of the patients hospitalized with COVID-19 in Spain are shown in Table 3. Although rates of respiratory insufficiency and ICU admission did not significantly differ among the two groups, the in-hospital mortality rate was significantly higher in SADs individuals (20% vs. 16%, *p* < 0.001).

In order to characterize the relationship between in-hospital mortality and SADs, a multivariate analysis of the whole population hospitalized with COVID-19 was performed (Table 4). Male sex (OR = 1.27, 95% CI 1.23–1.31), high blood pressure (OR = 1.11, 95% CI 1.07–1.15), obesity (OR = 1.23, 95% CI 1.17–1.30), the CCI (OR = 1.41, 95% CI 1.40–1.42) and respiratory insufficiency (OR = 2.92, 95% CI 2.21–2.37) were all independently associated with death. By contrast, SADs did not dive a raise to a higher mortality risk patients admitted with COVID-19 (OR = 0.93 95% CI 0.78–1.11). When individual SADs conditions were considered in the analysis, one by one (Figure 1), neither SLE (OR = 0.71, 95% CI 0.47–1.07), Sarcoidosis (OR = 1.17, 95% CI 0.73–1.89), SjS (OR = 0.81, 95% CI 0.57–1.17), SSc (OR = 1.31, 95% CI 0.78–2.16), IIM (OR = 0.96, 95% CI 0.26–3.58), MUCTD (OR = 1.20, 95% CI 0.52–2.77), BD (OR = 1.35, 95% CI 0.52–3.56) or SV (OR = 1.03, 95% CI 0.72–1.46) were independently related to death after adjustment.

Finally, the main prognostic factors for COVID-19 in the SADs population were evaluated in a multivariate analysis (Table 5). Only age (OR = 1.05, 95% CI 1.04–1.07), heart failure (OR = 1.67, 95% CI 1.10–2.49), chronic kidney disease (OR = 1.29, 95% CI 1.05–1.59) and liver disease (OR = 1.97–1.13–3.44) were found to be conditions independently related to death from COVID-19 in SAD patients.

## 4. Discussion

The impact of SADs on the outcome of patients hospitalized with COVID-19 during 2020 was evaluated from this large nation-wide registry. Our results revealed that the higher burden of comorbidities in these patients was responsible for their increased mortality rates. Therefore, none of the different SADs themselves gave rise to a higher COVID-19 mortality risk.

Several studies have assessed whether rheumatic diseases are associated with worse COVID-19 outcomes, with unclear results during the early pandemic period [18,19,20,21,22]. In subsequent waves, larger and multicentric cohorts confirmed that increasing age and pre-existing comorbidities were associated with higher severity and mortality in these patients. However, rheumatic disease was not an independent prognostic factor [23,24,25,26,27,28,29]. Finally, the OpenSAFELY study, and a more recent meta-analysis published in 2022, confirmed that, while rheumatic diseases might indeed be associated with COVID-19 death, their impact on mortality is overwhelmed by other conditions, such as obesity, respiratory disease, diabetes, severe kidney chronic disease, cancer and other immunosuppressive conditions [4,30]. In addition, these major studies raised the question of the vague definition and heterogeneity of these conditions in the different cohorts. Moreover, some of the aforementioned reports also found significant differences, in terms of disease course and outcomes, when chronic inflammatory arthritis was compared to other connective tissue diseases [20,26,28,31,32]. Therefore, it seems that certain conditions, such as RA, PM, PsA or other Spondylarthritis, which in fact represent the majority of the patients included in these registries, might not be equivalent to SLE, SjS, Sarcoidosis, SSc or SV, among others [4,20,22,23,24,25,28,29,31,32]. This is not surprising since the pathophysiology, systemic involvement, and impact on mortality among these diseases differ significantly [33,34,35]. Accordingly, we only analyzed in our national registry systemic rheumatic or autoimmune diseases, excluding chronic inflammatory arthritis, in order to understand their real significance in COVID-19 outcomes.

In the present study, we identified that patients with SADs presented similar respiratory insufficiency but higher mortality during hospitalization due to COVID-19 than the overall population. These differences might be accounted for by the higher number of comorbidities known to predispose to severe COVID-19, as confirmed by the multivariate analysis and identified in other populations with inflammatory mediated diseases including RA, PM or PsA [23,24,31,32]. Thus, SADs patients in our registry were admitted with similar respiratory insufficiency and COVID-19 severity, but due to their baseline conditions, their clinical outcome was significantly worse. In addition, we confirmed in one of the largest studies exploring COVID-19 outcomes in patients with rheumatic diseases, that neither SLE, SjS, Sarcoidosis, SSc, IIM, MUCTD BD or SV, individually, were associated with a higher mortality risk. Whilst previous reports have analyzed the impact of SLE or Sarcoidosis with similar conclusions, we identified that some of these diseases, less studied due to their lower prevalence, do not relate to worse COVID-19 outcomes [35,36,37].

Our results highlight that COVID-19 prognosis in SADs patients is mainly related to prior organ damage, either due to direct tissue involvement, pharmacological toxicity or sustained inflammation, and not the underlying autoimmune disease itself. From this perspective, certain conditions, such as heart failure, liver disease and chronic kidney disease, typically related to disease damage or subsequent treatments, were more frequent in SADs patients than in the general population and were, in turn, the main determinants of mortality in SAD patients in our study. Therefore, we believe that these findings strengthen the view that the best strategy to avoid fatal outcomes from COVID-19 in this population requires that first, in order to avoid chronic and irreversible organ damage, there is prompt and effective treatment of flares, with less toxic drugs and schemes, as a priority in these patients [16,38]. Second, vaccination policies and regimes must be individualized for individuals with SADs, taking into account that these patients must be at higher risk, even at a younger age, due to a higher rate of baseline comorbidities and prior chronic organ damage [39].

Finally, COVID-19 prognosis in the SADs population might not only be determined by baseline conditions, since immunosuppression itself may play a key role. While there is conflicting evidence regarding the impact of some treatments, to date there is solid data supporting the fact that certain drugs, mostly glucocorticoids and Rituximab, are associated with worse COVID-19 outcomes, both in the general population and in patients with SADs [8,23,24,26,27,31,40]. Unfortunately, we were not able to consider this issue, since no information regarding treatments could be retrieved from the database. However, it is not difficult to conjecture, based on previous evidence, the effect that the combination of these treatments might have over a SAD patient, with the burden of comorbidities as described in our cohort, who is exposed to SARS-CoV-2.

Our study has several limitations. Whilst the SNRHD records hospital discharge diagnoses along with demographic data, this database lacks information related to previous clinical conditions, such as treatment or disease status. Therefore, we have not been able to determine the impact of disease activity and immunosuppressive drugs on COVID-19. In parallel, we were not able to precisely define COVID-19 severity considering clinical and laboratory parameters. However, we believe that the lack of this data is compensated by the size of the study population, the nation-wide spectrum of the study population and the statistical power of the analysis. Secondly, we only analyzed the year 2020 so as to avoid selection or information bias regarding vaccination in this population. However, nowadays, the majority of the population in Spain has received the complete vaccine cycle and new variants have been discovered [41]. Therefore, further studies are needed to evaluate the impact of the vaccine in this population, as well as COVID-19 outcomes in SAD patients after vaccination.

## 5. Conclusions

In conclusion, the COVID-19 mortality rate in patients with SADs in Spain during 2020 was 20%, significantly higher than other hospitalized patients. It was related to the higher burden of comorbidities, probably secondary to direct organ damage and sequelae of their condition, and not due to SADs themselves. Therefore, efforts should be focused on the optimal management of SADs, minimizing the impact of organ damage that has been shown to determine COVID-19 prognosis.

## Figures and Tables

**Figure 1 viruses-14-01631-f001:**
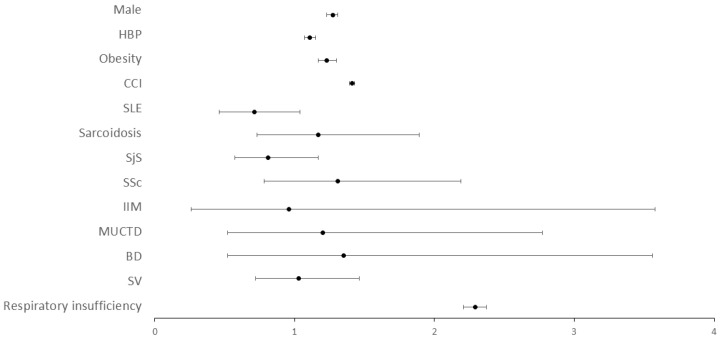
Risk factors for mortality in COVID-19 hospitalized patients for each SAD. The figure shows the COVID-19 risk factors for mortality in the Spanish population in 2020, considering the different systemic autoimmune disease. A binary logistic regression analysis was performed for each SAD. HBP: High blood pressure, CCI: Charlson Co-morbidity Index, SLE: Systemic Lupus Erythemathosus, SjS: Sjogren’s Syndrome, SSc: Systemic Sclerosis, IIM: Idiopathic Inflammatory Myopathies, MUCTD: Mixed and Undifferentiated Connective Tissue Disease, BD: Behçet’s Disease, SV: Systemic Vasculitides.

**Table 1 viruses-14-01631-t001:** Main demographics of the COVID-19 hospitalized patients.

	Total N (%)	SADs N (%)	Non-SADs N (%)	*p*-Value
COVID-19 hospitalized patients	117,694	892 (0.8)	116,802 (99.2)	
Mean age (mean, SD)	66.5 (18)	67.5 (15.6)	66.5 (18)	0.055
Male sex N (%)	66,685 (57)	257 (29)	66,428 (57)	<0.001
Ethnicity				
Caucasian	85,977 (73)	667 (75)	85,310 (73)	0.256
Arabic	2786 (2)	25 (3)	2761 (2)	0.380
Black	1592 ()	13 (2)	1579 (1)	0.770
Asian	387 (0.3)	0	387 (0.3)	0.129
Latin-American	9296 (8)	54 (6)	9242 (8)	0.043
Hindu	156 (0.1)	2 (0.2)	154 (0.1)	0.331
Unknown	17,500 (15)	131 (15)	17,369 (15)	0.920

SADs: Systemic Autoimmune Diseases, SD: Standard deviation.

**Table 2 viruses-14-01631-t002:** Distribution of major baseline comorbidities of the COVID-19 hospitalized patients.

	Total N (%)	SADs N (%)	Non-SADs N (%)	*p*-Value
High blood pressure	56,701 (48)	445 (50)	56,256 (48)	0.313
Diabetes mellitus	28,094 (24)	184 (21)	27,910 (24)	0.023
Uncomplicated	18,595 (16)	114 (13)	18,481 (16)	0.01
End-organ damage	9499 (8)	70 (8)	9429 (8)	0.847
Obesity	13,966 (12)	101 (11)	13,865 (12)	0.674
Ischemic heart disease	7858 (7)	56 (6)	7802 (7)	0.680
Heart failure	14,199 (12)	163 (18)	14,036 (12)	<0.001
Peripheral vascular disease	5059 (4)	68 (8)	4991 (4)	<0.001
CVA or TIA	7308 (6)	58 (7)	7250 (6)	0.683
Hemiplejia	1717 (2)	11 (1)	1706 (2)	0.675
Dementia	10,146 (9)	61 (7)	10,085 (9)	0.066
Chronic lung disease	16,814 (14)	206 (23)	16,608 (14)	<0.001
Peptic ulcer disease	341 (0.3)	3 (0.3)	338 (0.3)	0.747
Liver disease	6001 (5)	74 (8)	5927 (5)	<0.001
Mild	4065 (4)	61 (7)	4867 (4)	<0.001
Moderate to severe	1073 (1)	13 (2)	1060 (1)	0.106
Chronic kidney disease	13,232 (11)	160 (18)	13,072 (11)	<0.001
Localized solid tumor	433 (0.4)	3 (0.3)	430 (0.4)	1
Metastatic solid tumor	701 (0.6)	3 (0.3)	698 (0.6)	0.506
Leukemia	697 (0.6)	4 (0.4)	693 (0.6)	0.825
Lymphoma	610 (0.5)	3 (0.3)	607 (0.5)	0.638
HIV	234 (0.2)	2 (0.2)	232 (0.2)	0.699
CCI (mean, SD)	3.5 (2.6)	3.9 (2.4)	3.5 (2.6)	<0.001

SADs: Systemic Autoimmune Diseases, CVA: Cerebrovascular accident, TIA: Transient ischemic attack, HIV: human immunodeficiency virus, CCI: Charlson Co-morbidity Index.

**Table 3 viruses-14-01631-t003:** Clinical outcomes of the COVID-19 hospitalized patients according to SAD status.

	Total N (%)	SADs N (%)	Non-SADs N (%)	*p*-Value
Respiratory insufficiency	47,529 (40)	381 (43)	47,148 (40)	0.160
ICU admission	11,449 (10)	98 (11)	11,351 (10)	0.211
Admission length	10.6 (11.7)	11.2 (11.6)	10.6 (11.7)	0.09
ICU admission length	15.6 (17.6)	12.3 (10.5)	15.6 (17.6)	0.008
In-hospital mortality	18,858 (16)	174 (20)	18,864 (16)	0.05

SADs: Systemic Autoimmune Diseases, ICU: Intensive Care Unit.

**Table 4 viruses-14-01631-t004:** Risk factors for mortality in COVID-19 hospitalized patients.

	OR (95% CI)	*p*-Value
Male sex	1.27 (1.23–1.31)	<0.001
High blood pressure	1.11 (1.07–1.15)	<0.001
Obesity	1.23 (1.17–1.30)	<0.001
CCI	1.41 (1.40–1.42)	<0.001
SAD	0.93 (0.78–1.11)	0.433
Respiratory insufficiency	2.92 (2.21–2.37)	<0.001

CCI: Charlson comorbidity index, SAD: Systemic autoimmune disease.

**Table 5 viruses-14-01631-t005:** Risk factors for mortality in COVID-19 hospitalized patients with SADs.

	OR (95% CI)	*p*-Value
Age	1.05 (1.04–1.07)	<0.001
Heart failure	1.67 (1.10–2.49)	0.016
Chronic kidney disease	1.29 (1.05–1.59)	0.015
Liver disease	1.97 (1.13–3.44)	0.018

CCI: Charlson comorbidity index, SADs: Systemic Autoimmune Diseases.

## Data Availability

The data presented in this study are available on request from the corresponding author. The data are not publicly available since the database analysis has to be approved by the Spanish Ministry of Health.

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
