# Peer review of "Systemic Autoimmune Diseases in Patients Hospitalized with COVID-19 in Spain: A Nation-Wide Registry Study"

_viruses, 2022, doi:10.3390/v14081631_

Round 1
Reviewer 1 Report
Overall, I believe this is a well-designed study and will make a valuable contribution to the current literature. The overall structure of the manuscript is satisfactory. My comments below primarily pertain to further clarification or syntax.
Lines 18-21: This language does not make clear whether patients had these syndromes before contracting COVID vs. developing these syndromes after contracting COVID. Please define explicitly in the abstract.
Lines 22-25: was there a time point used for mortality? What was average length of follow-up?
Line 24-25: why are there no confidence intervals?
Line 36-37: unprecedent should be "unprecedented". Also what is the global agenda? This statement assumes there is some kind of unified global agenda to begin with.
Line 37-38: 6 million deaths due to COVID. Much more than 6 million deaths occurred in 2020-2021 if considering all-cause mortality.
Line 40: heterogenicity is used when discussing alleles of a gene. I believe you want to use "heterogeneity" here. You used heterogeneity everywhere else in the manuscript.
Lines 45-48: certain immunosuppressants, not all immunosuppressants. Mycophenolate is a predictor of severe COVID-19 in liver transplant patients (doi: 10.1016/j.jhep.2020.07.040). I would rephrase this sentence as: “The proven efficacy of certain immunosuppressants, such as glucocorticoids and IL-6 inhibitors (e.g. tocilizumab), especially during the hyperinflammatory phase of the disease, suggests that immunosuppression could provide protection against COVID-19.”
Lines 48-49: please specify exactly which patients are considered “immunocompromised” in this study. Also, the reference you used has a limitation that was not addressed by the authors of the review: this review included only 110 patients, mostly cancer and transplant patients, who tend to have closer medical care/scrutiny, better follow-up/medication adherence/laboratory monitoring than the general population. Is their immunocompromised status truly responsible for their better outcomes or is it because they receive better care due to their immunocompromised condition?
Line 54-56: rephrase as “Systemic autoimmune diseases (SAD) is comprised of a group of heterogeneous disorders in which persistent immune activation leads to a chronic inflammatory environment and direct tissue injury”
Line 61: the “disease itself” is ambiguous here. Are you referring to COVID-19 as the disease or are you referring to the underlying autoimmune disease? Please specify.
Lines 71-80: is this language necessary? Lines 74-80 describe practices that are pretty much universally accepted.
Line 89-91: rephrase as” Baseline conditions, including the presence of respiratory insufficiency, were recorded… “
Lines 97-103: like my first comment, this language does not make clear whether patients had these conditions prior to developing COVID or if they developed it during their hospitalization with COVID. Please explicitly define.
Line 111: rephrase as “We analyzed the overall SAD patient population and performed stratified analyses for each individual SAD as well.”
Lines 128-129: does your national database have an estimate for how many in your country have SAD? I’m curious if patients with SAD were more likely to be hospitalized due to COVID vs. those without.
Line 147: was there a time point associated with mortality?
Line 183: grave should be “gave”
Line 189: rephrase as “…however, rheumatic disease was not an independent prognostic factor”
Lines 203-206: this should be specified in the methods
Line 222: you mean the underlying autoimmune disease, correct?
Lines 234-237: please clarify. It is unclear whether you are talking about outcomes in the context of COVID or outcomes in the context of the underlying disease (ie SAD). Based on your references, it appears that you are talking about patients with SAD who developed COVID.
Line 246: this is a rather significant limitation. I believe this is significant enough to mention in the abstract.
Author Response
Overall, I believe this is a well-designed study and will make a valuable contribution to the current literature. The overall structure of the manuscript is satisfactory. My comments below primarily pertain to further clarification or syntax.
Thank you for your review. All comments have been deeply considered, and answered as follows.
Lines 18-21: This language does not make clear whether patients had these syndromes before contracting COVID vs. developing these syndromes after contracting COVID. Please define explicitly in the abstract.
Thank you for your suggestion. It has been re-written in order to concrete that these conditions were present before contracting COVID-19.
Lines 22-25: was there a time point used for mortality? What was average length of follow-up?
Thank you, we have concreted deaths as in-hospital mortality. Follow-up was discontinued after discharge, according to the database structure.
Line 24-25: why are there no confidence intervals?
In the initial version we did not include confidence intervals according to the abstract word-limit. This has been corrected.
Line 36-37: unprecedent should be "unprecedented". Also what is the global agenda? This statement assumes there is some kind of unified global agenda to begin with.
Thank you for your comment. This has been rewritten: The surge of SARS-CoV-2 infections in China towards the end of 2019 changed the global context in an unprecedented way.
Line 37-38: 6 million deaths due to COVID. Much more than 6 million deaths occurred in 2020-2021 if considering all-cause mortality.
We agree with the reviewer, and we have corrected the number accordingly (18 million deaths as measured by excess mortality vs 6 million reported deaths): In the two years following the onset of the pandemic, more than 18 million deaths have been confirmed globally.
Line 40: heterogenicity is used when discussing alleles of a gene. I believe you want to use "heterogeneity" here. You used heterogeneity everywhere else in the manuscript.
Thank you very much for your comment. This has been corrected in the manuscript.
Lines 45-48: certain immunosuppressants, not all immunosuppressants. Mycophenolate is a predictor of severe COVID-19 in liver transplant patients (doi: 10.1016/j.jhep.2020.07.040). I would rephrase this sentence as: “The proven efficacy of certain immunosuppressants, such as glucocorticoids and IL-6 inhibitors (e.g. tocilizumab), especially during the hyperinflammatory phase of the disease, suggests that immunosuppression could provide protection against COVID-19.”
We really appreciate your suggestion. The sentence has been rephrased accordingly.
Lines 48-49: please specify exactly which patients are considered “immunocompromised” in this study. Also, the reference you used has a limitation that was not addressed by the authors of the review: this review included only 110 patients, mostly cancer and transplant patients, who tend to have closer medical care/scrutiny, better follow-up/medication adherence/laboratory monitoring than the general population. Is their immunocompromised status truly responsible for their better outcomes or is it because they receive better care due to their immunocompromised condition?
In the lines you commented, we explained the controversy regarding immunosuppression in this population, but, as mentioned in the methods and limitations section, we could not concrete which patients were receiving immunosuppressants drugs. Therefore, we cannot consider immunocompromise in the analysis. Considering the second question, we fully agree with the fact that the better care these patients were receiving before or during COVID-19 might impact their outcome. However, this has not been always proven in the literature. Therefore, we consider that this reference might help to explain this conflicting results.
Line 54-56: rephrase as “Systemic autoimmune diseases (SAD) is comprised of a group of heterogeneous disorders in which persistent immune activation leads to a chronic inflammatory environment and direct tissue injury”
The sentence has been re-written has suggested. Thank you.
Line 61: the “disease itself” is ambiguous here. Are you referring to COVID-19 as the disease or are you referring to the underlying autoimmune disease? Please specify.
Certainly, ‘disease itself’ was not clear enough. This has been checked: Therefore, COVID-19 prognosis in this population might be determined by organ damage, immunosuppression and the underlying autoimmune disease itself.
Lines 71-80: is this language necessary? Lines 74-80 describe practices that are pretty much universally accepted.
In our opinion, we believe that the first two sentences highlight the strength and the utility of the database analyzed. On the other hand, lines 74-80 were indeed misplaced from the template, and have been deleted. Thank you.
Line 89-91: rephrase as” Baseline conditions, including the presence of respiratory insufficiency, were recorded… “
Thank you for your suggestion. The sentence has been rephrased.
Lines 97-103: like my first comment, this language does not make clear whether patients had these conditions prior to developing COVID or if they developed it during their hospitalization with COVID. Please explicitly define.
Thank you for your suggestion. Again, the sentence has been re-written in order to concrete that these conditions were present before contracting COVID-19.
Line 111: rephrase as “We analyzed the overall SAD patient population and performed stratified analyses for each individual SAD as well.”
Thank you for your recommendation.
Lines 128-129: does your national database have an estimate for how many in your country have SAD? I’m curious if patients with SAD were more likely to be hospitalized due to COVID vs. those without.
Unfortunately, the database cannot estimate the prevalence of SAD in the Spanish population. Indeed, this data would allow to clarify if the hospitalization risk significantly varies among SAD and non-SAD patients.
Line 147: was there a time point associated with mortality?
I am not sure I understand the question. Regarding pandemic time period, we did not consider the different waves or stages in the analysis. If you are asking about time point since symptom onset, we were not able to retrieve this information in the database. Finally, it should be noted that time from admission to death did not differed among the two groups.
Line 183: grave should be “gave”
This has been corrected. Thank you.
Line 189: rephrase as “…however, rheumatic disease was not an independent prognostic factor”
The sentence has been rephrased as suggested. Thank you.
Lines 203-206: this should be specified in the methods
We agree with the reviewer. This has been also specified in the methods section.
Line 222: you mean the underlying autoimmune disease, correct?
Indeed. This has been specified.
Lines 234-237: please clarify. It is unclear whether you are talking about outcomes in the context of COVID or outcomes in the context of the underlying disease (ie SAD). Based on your references, it appears that you are talking about patients with SAD who developed COVID.
Yes, we were talking about COVID-19 outcomes in SAD patients. As mentioned, this has been clarified: Finally, COVID-19 prognosis in SAD population might not only be determined by baseline conditions since immunosuppression itself may play a key role. While there is conflicting evidence regarding the impact of some treatments, to date there is solid data supporting that certain drugs, mostly glucocorticoids and Rituximab, are associated with worse COVID-19 outcomes both in the general population and in patients with SAD.
Line 246: this is a rather significant limitation. I believe this is significant enough to mention in the abstract.
Thank you for your suggestion. Since we believe that it is not appropriate to include a limitation in the abstract, we have mentioned it as follows: Whilst further studies should evaluate the impact of immunosuppression on COVID-19 outcomes in this population, efforts should be focused on the optimal management of SAD to minimize the impact of the organ damage that has been shown to determine COVID-19 prognosis.
Reviewer 2 Report
In this paper, the authors aimed to evaluate the clinical outcome of SAD patients hospitalized with COVID-19. This is a nationwide, retrospective and observational analysis of the patients admitted during 2020. The mortality rate was higher in SAD individuals (20% vs 16%, p<0.001). After adjustment by baseline conditions, SAD were not associated with a higher mortality risk (OR=0.93, 95%CI 0.78-1.11). Mortality in the SAD patients was determined by age (OR=1.05), heart failure (OR=1.67), LD (OR=1.97) and chronic kidney disease (OR=1.29). They conclude that the higher COVID-19 mortality rate seen in SAD patients hospitalized in Spain in 2020 was related to the higher burden of comorbidities, secondary to direct organ damage and sequelae of their condition.
The paper is well-structured and the study is well-design. This study formally demonstrates the impact of comorbidities in the mortality of COVID-19 patients.
We endorse the publication of the paper as is.
Minor point: please delete lines 74-80
Author Response
In this paper, the authors aimed to evaluate the clinical outcome of SAD patients hospitalized with COVID-19. This is a nationwide, retrospective and observational analysis of the patients admitted during 2020. The mortality rate was higher in SAD individuals (20% vs 16%, p<0.001). After adjustment by baseline conditions, SAD were not associated with a higher mortality risk (OR=0.93, 95%CI 0.78-1.11). Mortality in the SAD patients was determined by age (OR=1.05), heart failure (OR=1.67), LD (OR=1.97) and chronic kidney disease (OR=1.29). They conclude that the higher COVID-19 mortality rate seen in SAD patients hospitalized in Spain in 2020 was related to the higher burden of comorbidities, secondary to direct organ damage and sequelae of their condition.
The paper is well-structured and the study is well-design. This study formally demonstrates the impact of comorbidities in the mortality of COVID-19 patients.
We endorse the publication of the paper as is.
Minor point: please delete lines 74-80
Thank you for your review. As suggested, the lines 74-80 (misplaced from the template) have been deleted.